# Does the Argentine Tango Sustainably Improve Cancer-Associated Fatigue and Quality of Life in Breast Cancer Survivors?

**DOI:** 10.3390/cancers15235678

**Published:** 2023-11-30

**Authors:** Shiao Li Oei, Anja Thronicke, Jessica Groß, Thomas Rieser, Sarah Becker, Patricia Grabowski, Gerrit Grieb, Harald Matthes, Friedemann Schad

**Affiliations:** 1Research Institute Havelhöhe, Hospital Gemeinschaftskrankenhaus Havelhöhe, 14089 Berlin, Germany; anja.thronicke@havelhoehe.de (A.T.); thomas.rieser@havelhoehe.de (T.R.); harald.matthes@havelhoehe.de (H.M.); fschad@havelhoehe.de (F.S.); 2Breast Cancer Centre, Hospital Gemeinschaftskrankenhaus Havelhöhe, 14089 Berlin, Germany; jessica.gross@havelhoehe.de (J.G.); sarah.becker@havelhoehe.de (S.B.); 3Institute for Social Medicine, Epidemiology and Health Economics, Charité—Universitätsmedizin Berlin, Corporate Member of Freie Universität Berlin, Humboldt-Universität zu Berlin, and Berlin Institute of Health, 10117 Berlin, Germany; 4Interdisciplinary Oncology and Palliative Cancer Medicine, Hospital Gemeinschaftskrankenhaus Havelhöhe, 14089 Berlin, Germany; p.grabowski@havelhoehe.de; 5Institute for Medical Immunology, Charité—Universitätsmedizin Berlin, Corporate Member of Freie Universität Berlin, Humboldt-Universität zu Berlin, and Berlin Institute of Health, 10117 Berlin, Germany; 6Department of Plastic Surgery and Hand Surgery, Hospital Gemeinschaftskrankenhaus Havelhöhe, 14089 Berlin, Germany; gerritchristian.grieb@havelhoehe.de; 7Day Clinic for Internal Medicine/Apheresis Centre, Hospital Gemeinschaftskrankenhaus Havelhöhe, 14089 Berlin, Germany

**Keywords:** breast cancer, dance, integrative oncology, fatigue, physical activities, quality of life, supportive care, tango Argentino

## Abstract

**Simple Summary:**

Chronic cancer-related fatigue is a critical problem for breast cancer survivors and effective and sustained management is a key for improvement. A six-week tango Argentino program showed to be effective in reducing fatigue and improving quality of life, here we examined the sustainability of this effect over a twelve month period. It was observed that in the tango cohort, breast cancer survivors experienced a sustained improvement in physical functioning and a reduction in fatigue symptoms for at least six months, but not in a control cohort. Tango may be an appropriate activity to promote quality of life and also eventually to improve long-term clinical outcomes of cancer survivors.

**Abstract:**

Background: Chronic cancer-related fatigue is difficult to manage in breast cancer survivors. The tango trial showed that a six-week tango Argentino program was effective in reducing fatigue and improving quality of life, and here we investigated the sustainability of this tango program for breast cancer survivors. Methods: Stage I–III breast cancer survivors with increased fatigue symptoms were analyzed. The fifty participants in the tango trial were compared with a control cohort (*n* = 108) who did not participate in the tango program. Using the European Organization for Research and Treatment of Cancer Questionnaire C30 (EORTC-QLQ-C30) and the German version of the cancer fatigue scale (CFS-D) self-reported quality of life parameters were assessed and longitudinal changes, correlations, and association factors were calculated. Results: Significant improvements in fatigue (*p* = 0.006), physical functioning (*p* = 0.01), and diarrhea (*p* = 0.04) persisted in the 50 Tango participants at 6 months, but not in the control cohort. Twelve months after joining the tango program, increased fatigue was associated with reduced sporting activities (*p* = 0.0005), but this was not the case for tango dancing. Conclusions: The present results suggest that tango may be appropriate as a component of early supportive and follow-up care programs, to promote health-related quality of life and physical activity and also eventually to improve long-term clinical outcomes of breast cancer survivors. Trial registration: Trial registration numbers DRKS00013335 on 27 November 2017 and DRKS00021601 on 21 August 2020 retrospectively registered.

## 1. Introduction

Breast cancer continues to be the most common form of cancer among women worldwide [1]. Approximately one-third of breast cancer survivors experience moderate to severe symptoms of cancer-associated fatigue, which is a multidimensional syndrome, and has a profound negative impact on health-related quality of life (QoL) [2]. The risk of severe fatigue is relatively highest in patients who are treated with a combination of surgery, chemotherapy, radiation therapy, and hormonal therapy [3]. Symptoms of fatigue in patients with early breast cancer after surgery, chemotherapy, and follow-up care are usually experienced in a timely manner and often persist for many years afterwards [2,4]. Based on current scientific evidence, international practice guidelines recommend predominantly non-pharmacological interventions for the treatment of fatigue [5,6]. Mindfulness-based stress reductions, such as meditation and yoga in particular, have been shown to be efficacious to reduce fatigue levels [7] and from systematic reviews was concluded that QoL benefits can be reached with exercises, physical self-management, and cognitive-behavioral therapies [8]. A meta-analysis revealed that a supervised combination of resistance endurance with aerobic training is the most effective physical exercise to reduce fatigue in breast cancer patients [9]. International multidisciplinary roundtables concluded that cancer survivors can safely engage in sufficient exercise training to improve their QoL, alleviate cancer-associated fatigue, and currently, it is widely accepted that exercises, moderate resistance endurance training and physical activities are proposed as an add-on first-line intervention [10]. However, fatigue itself, low motivation, and lack of time, are often barriers to physical activity [11]. It also has been found that music therapy can exert positive effects on anxiety, fatigue, and QoL, thus may improve psychological and physical outcomes in cancer patients as well [12]. Meta-analyses have shown that the clinical effectiveness of music-based interventions is still unclear, but they appear to have a positive effect on the stimulation of overall motivation [13]. Unlike traditional exercise therapies, dance uniquely combines social, cognitive, and fitness components, and may be an appropriate approach to managing fatigue symptoms and improve QoL [14]. Thus, music, improvisation, and communication activities could have a stimulating effect and also promote physical activity. The first results from controlled clinical trials of belly dance or creative dance interventions for cancer survivors and effects on patient-reported outcomes have been published recently [15,16]. Unlike other dance styles, such as foxtrot and waltz, which offer little rhythmic variety, the Argentine tango includes rhythmic variations and allows improvisation with spontaneous reactions, steps, and movements to the music. The Argentine tango may strengthen patients’ social network and also improve self-esteem and a systematic review on Argentine tango in Parkinson’s disease reported beneficial effects on fatigue and QoL [17]. A feasibility study of Argentine tango for cancer survivors was reported [18]. Our recently presented tango study in breast cancer patients one to three years after diagnosis demonstrated the efficacy of a six-week Argentine tango program on cancer-associated fatigue and quality of life [19]. The aim of the present study is to assess the sustainability of this Tango program for breast patients on fatigue and quality of life. In addition, with follow-up surveys the course of physical functioning and physical activity will also be part of this evaluation.

## 2. Materials and Methods

Here, we conducted a real-world data (RWD) study by longitudinal evaluation of the tango trial [19] and the oncological patient registry data from Network Oncology (NO) [20]. The tango trial was an investigator initiated trial, and the study protocol and results were published [19,21]. The NO study and the tango trial were approved by the responsible local ethics committee (Ärztekammer Berlin) and registered (Trial registration numbers DRKS00013335 and DRKS00021601).

### 2.1. Participants, Study Design, and Procedure

The NO is a multi-centre oncological register study to evaluate therapy concepts in health services research [20]. In addition, QoL questionnaires from cancer patients are evaluated in the NO. For the present study, primary breast cancer patients, aged between 18 and 100 years, who answered QoL questionnaires, and from whom written informed consent was obtained beforehand, were screened [22]. Demographic data, QoL questionnaires, as well as information on diagnosis, histology, type of surgery, and treatment received were documented and collected at different time points. For a control cohort, NO breast cancer survivors who were diagnosed with stage I to III breast cancer and reported increased fatigue symptoms (CFS-D total > 23) at the one-year follow-up were included. The participants of the tango trial were recruited between June 2020 and May 2022 according to the published study-protocol [21] and received the six-week tango program, as described in detail [19]. Demographic, behavioral, and medical information of the tango cohort were collected with clinical interviews, self-reported fatigue, and QoL questionnaires were surveyed prior to, after the intervention, and six, and twelve months thereafter.

### 2.2. Outcome Measures

Self-reported symptoms of fatigue and quality of life were assessed using German versions of the questionnaires: the cancer fatigue scale (CFS-D) [23] and the European Organization for Research and Treatment of Cancer Questionnaire C30 (EORTC-QLQ-C30) [24]. For the equations, calculation, and handling of missing data, the EORTC-QLQ-C30 Scoring Manual was followed. In addition, the participants in the tango program were also asked questions about their sports and dance activities in the follow-up surveys.

### 2.3. Statistical Analysis

For statistical analyses Excel 2010 (Microsoft, Redmond, WA, USA) and the R software (R Version 4.2.1, R core team, Vienna, Austria), a language and environment for statistical computing, were used. For longitudinal analyses mean differences were estimated and Student’s *t*-tests were performed to detect differences; *p*-values < 0.05 were considered significant. Adjusted multivariable linear regression analyses were performed to analyze associations between QoL scales and sport (no, 0; occasionally, 1; 1–2 weekly, 2; 2 < weekly, 3), dance (no, 0; occasionally, 1; regularly, 2), or online sport (no, 0; occasionally, 1; regularly, 2) activities, respectively. Adjustments were made for the variables of age and the menopausal status.

## 3. Results

Breast cancer survivors with elevated fatigue symptoms were enrolled in the tango trial an average of 1.7 years after their primary cancer diagnosis, and fifty completed the six-week Tango program [19] and were assessed with follow-up surveys at 6 and 12 months and compared to a Control cohort (Figure 1). For this Control cohort, a total of 278 primary breast cancer patients, cancer stage I-III, who were diagnosed and treated in breast cancer centers of the NO consortium between 2013 and 2021, who answered QoL questionnaires one and two years after their primary cancer diagnosis and who did not participate in the tango trial, were screened. At the one-year follow-up, 108 of these breast cancer survivors reported increased fatigue symptoms (CFS-D total > 23) (Figure 1). All baseline characteristics and the QoL scores before the tango program and at the one-year follow-up, respectively, are summarized in Table 1. There were no apparent differences in baseline characteristics between the tango and control cohorts (Table 1).

### 3.1. Comparison of QoL Outcomes in the Tango and the Control Cohort

A total of 50 (100%) participants from the tango group answered the six-month follow-up survey, and 49 (98%) answered the twelve-month follow-up survey. The longitudinal changes of the patient-reported QoL outcomes from the tango participants and the 108 breast cancer survivors in the control cohort were evaluated.

Improvements on some of the EORTC QLQ-C30 scales were reported at 6 and 12 months in the tango cohort and at 12 months in the control cohort (Table 2). Specifically, there was still a 5% improvement in physical functioning at 6 months (*p* = 0.01) and 12 months (*p* = 0.02) in the tango cohort. Fatigue (*p* = 0.006) and diarrhea (*p* = 0.04) were also reduced by 7% after 6 months, while insomnia was not. Figure 2 shows the time course of physical functioning and symptoms of fatigue, insomnia, and diarrhea for the tango and control cohorts.

### 3.2. Follow-Up Evaluation of the Tango Cohort

There were no known cases of any of the fifty participants in the tango trial who did not survive the 12-month follow-up period. Between the 6th and 12th month of follow-up, an additional cancer diagnosis was reported in 3 (6%) participants (breast cancer diagnosis on the other side, local cancer recurrence, or occurrence of bone metastases), and 4 (8%) participants self-reported secondary diseases (pneumonia; depression; fractured ribs; psychological disorder). Participants in the tango trial were surveyed about their activities and preferences in sports and dance at six months and again at twelve months (Table 3).

From Table 3, a trend toward less sports activities at 12 months than at 6 months (χ^2^ = 3.52; Cohen’s D *d* [95%CI] = 0.38 [−0.03, 0.79]) can be inferred. In the 6-month survey, a free-text question was asked, “Did you like dancing in a group? Or did you prefer dancing alone with the instructor?” 92% of the Tango cohort experienced tango as a good group program, 4% would have preferred individual instructions (Table 3) and 20% of the participants explicitly stated that they enjoyed the tango group classes a lot. The majority (64%) would have liked to take a regular guided dance class, 54% of participants reported practicing tango at home during the 6 months, 61% performed dance activities in the following months, while 37% did not practice any kind of dance exercise at all (Table 3). In addition, 30% of the participants reported an interest in online offers in the 6-month survey, and the majority (69%) reported never having used online offerings for sports or dance practice at the 12-month survey (Table 3).

Associations between QoL scales and physical activities were analyzed using adjusted multivariate analyses. The estimates for the CFS-D scales and subscales and a selection of EORTC QLQ C30 scales twelve months after the tango classes are summarized in Table 4.

Specifically, an association between high affective fatigue and lower sports engagement was found 12 months after Tango (estimate β = −0.034; *p* = 0.0005) and likewise regarding online sports (estimate β = −0.01; *p* = 0.01). Furthermore, increases in global health status (estimate β = 0.03; *p* = 0.002) and role functioning (estimate β = 0.01; *p* = 0.03), as well as decreased pain (estimate β = −0.014; *p* = 0.004) were associated with increases in sports engagements, whereas no associations were found between any QoL scale and dance exercises (Table 4).

## 4. Discussion

The randomized, controlled tango trial in breast cancer survivors found superiority of the tango over wait-list controlled-for fatigue symptoms [19], and the follow-up studies presented here show sustained improvements, particularly in physical functioning and fatigue symptom reduction, for at least six months.

The key elements of dance movement therapy as a group intervention are behavioral engagement, the increase in bodily awareness, and the stimulation of creativity and imagination [25]. The effectiveness of dance movement interventions to improve psychological and physical outcomes is not limited to cancer, and a cumulative, positive impact on mental health, particularly for mood disorders, has been reviewed [25]. Interventions that provide individualized rehabilitation, taking into account patients’ individual needs and preferences, were reviewed and positive effects were seen for resistance training and psychosocial interventions such as cognitive behavioral therapy. Together, these were shown to have positive effects on QoL, anxiety, depression, and mood disturbance [26]. Nevertheless, trials of high methodological quality are still needed to conduct systematic meta-analyses on the effectiveness of dance movement therapies [27]. However, the movement elements of dance interventions cannot be easily quantified and standardized for comparison with other movement programs. On the contrary, possibilities and dimensions such as fostering teamwork and self-awareness, learning a sequence of steps in harmony with the music, etc. are not necessarily included in movement therapies, but are in dance. Dancing requires great receptivity and ability to adapt to your partner’s subtle changes in balance, timing, speed, and direction. Interestingly, a mathematical–statistical calculation and description of tango dancing was recently published in this context [28], reflecting what variability is possible with tango.

As a common side effect of cancer therapy, approximately one-third of breast cancer survivors experience symptoms of diarrhea [29], but this burden is typically underestimated and does not receive special attention [30]. Specifically, a significant reduction in diarrhea symptoms was observed after the tango program [19], and from the follow-up surveys here it can be assumed that this effect lasts for approximately six months. Diarrhea secondary to cancer or its treatment may require dose reduction, modification, or discontinuation of effective therapies, which may adversely affect long-term clinical outcomes [30]. Although mild forms of diarrhea are not clinically relevant, diarrheal symptoms are also reported in long-term cancer survivors and have a major impact on QoL [31]. For cancer-associated diarrhea, guidelines suggest oral hydration, pharmacologic treatment options and dietary counselling [32]. Tango, as a non-pharmacological intervention, seems to contribute to the stabilization of the digestive system and the improvement of QoL.

Meanwhile, recent results from randomized controlled trials with breast cancer survivors show that dancing as an adjunctive therapy may improve QoL and decrease depression and pain perception [15,16]. It has long been known that mind–body therapies combined with physical activity and physical self-management interventions have beneficial effects on perceived quality of life, including fatigue symptoms [33,34]. Accordingly, exercises, yoga, relaxation techniques, music therapy, and mediation are suitable therapies for treating anxiety, stress, depression, and mood disorders and have been included in international guideline recommendations [5,6,35]. Dance should also be added to the list of effective therapies in these guidelines. The majority of participants also wanted to continue the tango exercises (Table 3), and an appropriate continuation of this program could lead to more lasting effects in terms of QoL including fatigue. In the present study, tango was almost universally appreciated as a group intervention. The tango program here consisted of six weekly 60 min sessions, so it could and should be expanded, and group tango classes should be offered on a recurring basis. Patients should also be guided to self-management and for this also eHealth tools could be useful [36]. However, in the follow-up survey of our tango cohort, only about one-third of respondents participated in online sports activities. A reduction of fatigue symptoms appeared to be associated with the intensity of sporting activity [19], which is consistent with the indisputable evidence that being physically active is supportive for QoL. But cancer survivors with severe fatigue symptoms are particularly inhibited from engaging in physical activity [11], and here we were able to show that twelve months after the tango program, it is precisely associative fatigue and pain that are negatively correlated with the practice of sports activities (Table 4). Severe fatigue is a barrier to exercise, but these patients need support and dance seems to be an appropriate way to achieve this. Figure 3 proposes a model of how fatigue and physical activities may interact. Immediately after tango physical activity and tango synergistically improve symptoms of fatigue [19]. This relationship changes over time, and after twelve months the relationship is reversed, with increasing symptoms of fatigue limiting sporting activities (Table 4). However, we found that twelve months after participation in the tango program, there was no negative correlation between dancing activity and fatigue (Table 4). Tango may, therefore, be a way out of this vicious circle and may be a motivator for physical activity in spite of fatigue.

Although adherence to the six-week tango program was good, dropouts had worse baseline fatigue and QoL scores [19]. Thus, they actually had a greater need for therapy and would be even more likely to drop out if they exercised longer. Therefore, a program must not be too ambitious, or it must be individually adaptable to each participant, which is possible with Argentine tango. Dancing also allows for face-to-face contact, social interaction, and experiencing new knowledge, both going forward and backward, which online sports activities cannot provide. When dancing the tango, nothing is known in advance, and a key feature of the Argentine tango is creativity and improvisation, which may awaken joy and inspiration in participants.

Some limitations of this study are that we had a non-blinded study design, no objective outcome measures, we did not have a direct parallel comparison group that did not receive tango, and detailed information on sleep quality and exercise was not available for this control cohort. In addition, our results presented here are limited to early stage breast cancer patients, as the study participants were predominantly stage I and II breast cancer patients. Furthermore, the fact that tango and most of the follow-up occurred during the COVID-19 pandemic limits generalizability. The strengths of our study are the high adherence despite the COVID-19 pandemic situation, and that adjusted multivariate regression analyses reduced confounding bias and allowed the identificatino of association factors between QoL scales and physical activity.

## 5. Conclusions

The present follow-up evaluation of the tango trial indicates a sustained efficacy of the Argentine tango program on quality of life, including fatigue, in breast cancer survivors. A number of interventions have been proposed that have been shown to be effective in treating cancer-associated fatigue. However, they are not suitable for all cancer survivors. A program that is not too physically demanding and can be tailored to individual needs can reach those who need the most support. As dancing activities were found not to interfere with any of the QoL scales, dancing may be a way to motivate physical activity despite fatigue. Dancing the tango is a way to develop body awareness and creativity in a group setting that simultaneously addresses cognitive, social, and emotional skills as well as individual passions. Therefore, as part of early supportive and follow-up care, it is necessary to develop outpatient rehabilitation programs that consist of multiple components to prevent fatigue, promote quality of life, and also eventually improve the long-term clinical outcomes of breast cancer survivors.

## Figures and Tables

**Figure 1 cancers-15-05678-f001:**
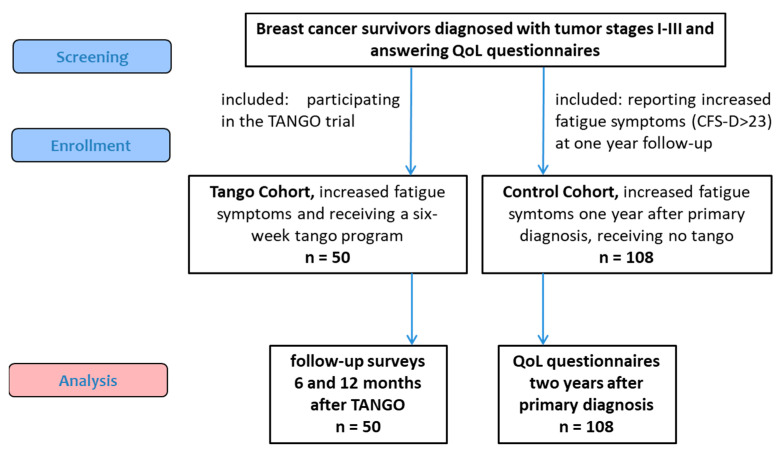
Flow chart of the study population. CFS-D: Cancer fatigue scale, German version; QoL: Quality of life.

**Figure 2 cancers-15-05678-f002:**
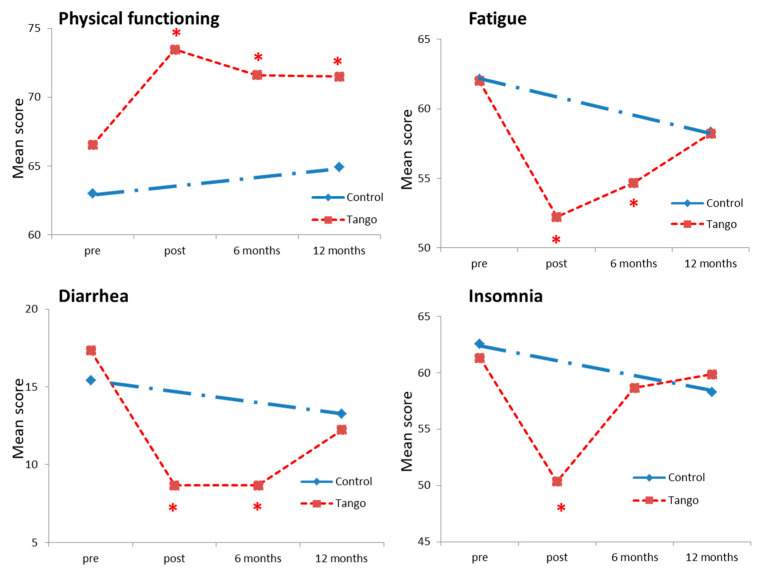
Time course of the EORTC QLQ-C30 sores for tango and control groups. The physical functioning, symptoms fatigue, diarrhea, and insomnia scores for the tango cohort (red, n = 50) pre, post tango and 6 and 12 months later and for the control cohort (blue, n = 108) are shown. Significant changes related to the scores prior intervention are indicated. *: *p*-value < 0.05.

**Figure 3 cancers-15-05678-f003:**
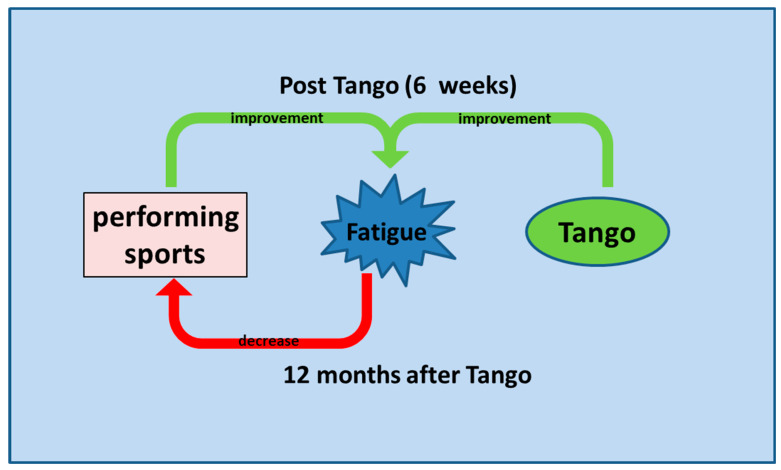
Fatigue and physical activity circle model.

**Table 1 cancers-15-05678-t001:** Baseline characteristics.

	Tango	Control
Number of patients, n (%)	50 (100)	108 (100)
Age, years, mean (SD)	61.3 (9.9)	61.6 (11.7)
UICC stages, n (%)		
I	23 (46)	51 (47)
II	20 (40)	48 (44)
III	7 (14)	9 (8)
Hormonal status, n (%)		
Premenopausal	2 (4)	7 (6)
Postmenopausal	48 (96)	101 (94)
Interventions received, n (%)		
Endocrine therapy	34 (68)	84 (78)
Chemotherapy	24 (48)	47 (44)
Immunotherapy	6 (12)	13 (12)
QoL questionnaires, mean (SD)		
CFS-D total	31.0 (9.3)	32.2 (6.3)
EORTC QLQ C30 global health	54.1 (16.9)	54.6 (17.7)
Physical functioning	66.5 (20.8)	63.0 (20.5)
Role functioning	54.0 (27.0)	49.4 (27.7)
Emotional functioning	42.1 (25.4)	42.4 (23.3)
Fatigue	62.0 (24.6)	62.2 (19.9)
Insomnia	61.3 (29.9)	62.6 (33.6)
Diarrhea	17.3 (29.2)	15.4 (25.8)

n: numbers; SD: standard deviation.

**Table 2 cancers-15-05678-t002:** Differences of QoL outcomes.

	Tango, at 6 Months n = 50 dmean (SD) *p*-Value	Tango, at 12 Months n = 49 dmean (SD) *p*-Value	Control n = 108 dmean (SD) *p*-Value
EORTC QLQ C30			
Global health	5.7 (18.4) 0.01 *	0.2 (17.5) 0.94	3.1 (18.6) 0.09
Physical functioning	5.1 (13.6) 0.01 *	4.8 (13.3) 0.015 *	2.3 (15.7) 0.24
Role functioning	3.7 (23.2) 0.27	0.0 (27.6) 1.0	5.3 (30.8) 0.13
Emotional functioning	7.0 (20.0) 0.018 *	1.4 (19.1) 0.67	6.7 (22.5) 0.003 *
Fatigue	−7.0 (17.9) 0.006 *	−3.7 (17.7) 0.15	−3.8 (20.7) 0.062
Insomnia	−2.7 (26.7) 0.49	−1.4 (22.6) 0.68	−4.1 (29.9) 0.16
Diarrhea	−8.7 (28.1) 0.036 *	−4.1 (23.0) 0.22	−2.2 (28.1) 0.4

EORTC QLQ C30: European Organization for Research and Treatment of Cancer Questionnaire C30; dmean: mean difference; n: numbers; SD: standard deviation. Significant *p*-values are indicated: *****: *p*-value < 0.05.

**Table 3 cancers-15-05678-t003:** Follow-up characteristics of the tango participants.

	At 6 Months, n = 50 n (%)	At 12 Months, n = 49 n (%)
	Did you engage in any sports activities?	
No = 0	10 (20)	12 (24)
occasionally = 1	3 (6)	7 (14)
1–2 weekly = 2	28 (56)	19 (39)
>2 weekly = 3	9 (18)	10 (20)
ND	0	1 (2)
	Did you like dancing in a group? Or did you prefer dancing alone with the instructor?	
preferred in the group	40 (80)	
no preference	6 (12)	
single, with instructor preferred	2 (4)	
ND	2 (4)	
	Would you like to take a regular guided dance class?	
no	4 (8)	
probably	14 (28)	
yes	32 (64)	
ND	0	
	Have you danced tango at home?	Did you do any kind of dance exercises?
No = 0	23 (46)	18 (37)
occasionally = 1	26 (52)	27 (55)
regularly = 2	1 (2)	3 (6)
ND	0	1 (2)
	Are you interested in online offers?	Have you used online offerings for sports or dance practice?
No = 0	18 (36)	34 (69)
probably/occasionally = 1	15 (30)	12 (24)
Yes/regularly = 2	15 (30)	2 (4)
ND	2 (4)	1 (2)

ND: not determined.

**Table 4 cancers-15-05678-t004:** Association factors between quality of life scales and physical activities.

CFS-D Fatigue	Any Sports Engagement?	Any Online Sports?	Any Dance Exercises?
total	−0.026 *	−0.008	0.0001
physical	−0.023 *	−0.003	0.0005
cognitive	−0.01	−0.006	0.001
affective	−0.034 **	−0.01 *	−0.001
**EORTC QLQ C30**	**Any Sports Engagement?**	**Any Online Sports?**	**Any Dance Exercises?**
global health status	0.027 **	0.007	0.002
physical functioning	0.018 ^+^	−0.002	0.006
role functioning	0.011 *	−0.004	−0.002
pain	−0.014 **	−0.001	−0.001

Adjusted for age and menopausal status, multivariate linear regression analyses were performed between the CFS-D (top panel) or EORTC QLQ-C30 (bottom panel) scales and physical activity 12 months after Tango. CFS-D scale and subscale estimates (top panel) were converted to percentages. Significant estimates and *p*-values are highlighted in bold and indicated: ******: *p*-value < 0.005; *****: *p*-value < 0.05; **^+^**: *p*-value < 0.01. CFS-D: Cancer fatigue scale, German version; EORTC QLQ C30: European Organization for Research and Treatment of Cancer Questionnaire C30.

## Data Availability

The datasets that support the findings in this article are not publicly available due to reasonable privacy and security concerns but are available from the corresponding authors upon reasonable request.

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
