# Peer review of "Does the Argentine Tango Sustainably Improve Cancer-Associated Fatigue and Quality of Life in Breast Cancer Survivors?"

_cancers, 2023, doi:10.3390/cancers15235678_

Round 1
Reviewer 1 Report
Comments and Suggestions for Authors
The downside of this interesting paper is the non-blinded design. Thus, in addition to confirming the advantages of Dance therapy with a larger population [as suggested by the authors] they should make it objective and valid by blinding the evaluation of the QoL, by persons who do not know if the patient was in the active/interventional or the control group.
Nevertheless, this paper is thought-provoking and the patient seems to enjoy this "treatment" without side effects!
Author Response
Response to Reviewer 1:
Thank you very much for your interest and your very appreciative remarks and comments.
In the limitations on page 10 lines 4-5, we have now added the non-blinded study design and the lack of objective outcome parameters.
All changes to the revised manuscript are presented in the track-change version of the document.
Reviewer 2 Report
Comments and Suggestions for Authors
The presented study has some advantages, such as demonstrated that a tango argentino program was effective in reducing fatigue and improving quality of life of breast cancer survivors, and also eventually to improve long-term clinical outcomes. The merit of the work is that the results were well structured, and the corresponding figures were clear and concise. However, I have some suggestions which must be taken into account so that the manuscript can be published.
In the Introduction section, it is necessary to explain the choice of tango argentino to study the effects on patients after breast cancer surgery. Why is it not discussed, for example, belly dancing or waltz? Is it so important to practice tango argentino or is any physical activity with communication between people within the group possible? Perhaps the author can reveal in more detail the mechanism of the effective action of the physical activity he chose for study. These questions must be answered in the manuscript.
It is important to discuss the topic of physical activity of patients after breast cancer surgery. Is a 6-week course justified for people after cancer surgery? How does this affect local and distant cancer recurrences? Why don’t the authors provide at least statistics on 3- and 5-year survival rates? If the authors do not have such statistics, they should provide and describe reliable papers demonstrating the safety and adequacy of this approach to patients' rehabilitation. Otherwise, it cannot be clinically justified and does not qualify for publication of the object material in Cancers. The target audience of such journal should see in the manuscript the clinical validity of such rehabilitation - the absence of a negative impact, determined by the criteria of patient survival and cancer recurrence.
Author Response
Response to Reviewer 2:
Thank you very much for your interest and your comments.
Now, in the introduction on page 2, lines 24 to 38, we have explained in detail the peculiarities and differences between tango and other dance styles.
No clear conclusions about the mechanism of action can be drawn here, but at most only some considerations. Corresponding additions now have been integrated in the discussion on page 8, lines 49 to page 9, line 25.
In this study, Tango is not examined as a rehabilitation program for breast cancer surgery. All participants received the Tango program approximately 1.7 years after their breast cancer surgery and outcome measures are not survival but cancer-associated fatigue and quality of life. Therefore, no survival or recurrence statistics should be reported in the follow-up data of the Tango study presented here. All changes to the revised manuscript are presented in a track-change version of the document.
Reviewer 3 Report
Comments and Suggestions for Authors
The title and content may give non-professional readers misleading guidance. Argentine tango can essentially be considered as part of a sport, it is difficult to say whether it is superior to other similar sports such as jogging or yoga. This article cannot answer this question. There is no report on the specific phase and quantity of the 50 enrolled patients, indicating that the survival of the enrolled subjects has no benefit for the results. I suggest rejecting this article directly.
Comments on the Quality of English LanguageOK
Author Response
Response to Reviewer 4:
Thank you very much for your interest and your remarks and comments.
In the introduction on page 2, lines 25 to 37, we have explained in detail the characteristics and differences between tango and other sporting activities or dance styles. From the results presented in Tables 3 and 4, some conclusions can be drawn regarding the differences between physical activities and tango, and the description of the relationships between fatigue and physical activity or tango in the discussion (page 9, lines 17-25) has now been revised.
The aim of the present study has now been better specified on page 2, lines 42-45. However, the endpoints measured in this study are not survival but cancer-related fatigue and quality of life, so survival rates cannot be addressed in this manuscript. All changes to the revised manuscript are presented in a track-change version of the document.
Round 2
Reviewer 2 Report
Comments and Suggestions for Authors
I agree with your edits to manuscript.
Author Response
We thank you for your agreement to the changes that we have made.
Reviewer 3 Report
Comments and Suggestions for Authors
The previously reviewed issues have been partially modified. The number of subjects in each period is related to the analysis results. According to the provided sample size list, there are only 7 and 9 subjects in Stage III for the experimental group and control group respectively. This means that the effect of this study on improving cancer-related fatigue and its applicability may be limited to Stage I to Stage II. This limitation should be stated in the article.
Comments on the Quality of English LanguageOK
Author Response
Thank you very much for your comments and suggestions. In accordance with your suggestions, the following sentence has now been added to the limitations on page 10:
In addition, our results presented here are limited to early-stage breast cancer patients, as the study participants were predominantly stage I and II breast cancer patients.
According changes to the revised manuscript are presented in the manuscript document in a track change mode.